# Sensitive HIV-1 DNA Pol Next-Generation Sequencing for the Characterisation of Archived Antiretroviral Drug Resistance

**DOI:** 10.3390/v15091811

**Published:** 2023-08-25

**Authors:** Johannes C. Botha, Matthew Byott, Moira J. Spyer, Paul R. Grant, Kathleen Gärtner, Wilson X. Chen, James Burton, Alasdair Bamford, Laura J. Waters, Carlo Giaquinto, Anna Turkova, Cindy L. Vavro, Eleni Nastouli

**Affiliations:** 1Great Ormond Street Institute of Child Health, University College London, London WC1N 1EH, UKeleni.nastouli@nhs.net (E.N.); 2Advanced Pathogen Diagnostics Unit, University College London Hospitals NHS Trust, London NW1 2PG, UK; 3Health Services Laboratories, London WC1H 9AX, UK; 4ViiV Healthcare, Durham, NC 27709, USA; 5Great Ormond Street Hospital for Children NHS Foundation Trust, London WC1N 3JH, UK; 6Medical Research Council Clinical Trials Unit, University College London, London WC1E 6BT, UK; 7Central and North West London NHS Foundation Trust, Mortimer Market, London WC1E 6JB, UK; 8Department of Women and Child Health, University of Padova, 35122 Padova, Italy; 9Fondazione Penta ETS, 35127 Padova, Italy

**Keywords:** HIV-1, HIV-1 DNA, drug resistance, provirus, NGS, sanger

## Abstract

Modern HIV-1 treatment effectively suppresses viral amplification in people living with HIV. However, the persistence of HIV-1 DNA as proviruses integrated into the human genome remains the main barrier to achieving a cure. Next-generation sequencing (NGS) offers increased sensitivity for characterising archived drug resistance mutations (DRMs) in HIV-1 DNA for improved treatment options. In this study, we present an ultra-sensitive targeted PCR assay coupled with NGS and a robust pipeline to characterise HIV-1 DNA DRMs from buffy coat samples. Our evaluation supports the use of this assay for Pan-HIV-1 analyses with reliable detection of DRMs across the HIV-1 Pol region. We propose this assay as a new valuable tool for monitoring archived HIV-1 drug resistance in virologically suppressed individuals, especially in clinical trials investigating novel therapeutic approaches.

## 1. Introduction

The ability of combination antiretroviral therapy (cART) to suppress HIV-1 replication safely and effectively has rendered it a manageable chronic condition. However, cART is not a cure and has no direct effect on the HIV-1 DNA reservoir [1,2,3]. Infectious proviruses present in long-lived cells comprise the stable HIV-1 DNA reservoir, which remains the main barrier to cure [4]. The development of drug resistance related to several factors, including poor adherence and provirus evolution [5,6,7,8], maintains the risk of viral rebound. Therefore, drug resistance testing in HIV-1 plasma RNA, before treatment initiation or in case of poor treatment response, remains an essential part of informed tailored therapy [1,9], but monitoring of archived/low percentages of drug resistance mutations (DRMs) on the DNA level is gaining significance for potential viral rebound despite ART.

Increased use of next-generation sequencing (NGS) for HIV-1 drug resistance (HIVDR) testing on HIV DNA offers enhanced sensitivity not obtainable by traditional Sanger sequencing [10,11,12]. Several DNA HIVDR NGS-based assays are available, with the most prominent clinically used assays reporting amplification success rates of 58.4% to 94% from purified peripheral blood mononuclear cells (PBMCs) [13,14]. Other studies reported HIV-1 amplification sensitivity as low as 10 and 131 DNA copies per reaction [15,16]. The clear benefit of NGS HIVDR genotyping supports its use in the clinical environment, although low-level resistance results should be interpreted with care [17,18]. The clinical use of total DNA HIVDR genotyping has gained interest for monitoring HIV-1 DNA stability in people with suppressed viral loads [1,19]. HIVDR genotyping in DNA could prove useful for people that are virologically suppressed and require regimen switching, preliminary investigation of clonal viremia or for monitoring HIV-1 DNA stability (archived resistance) in clinical trials investigating novel therapeutic approaches [19,20,21].

In this study, we present an ultra-sensitive assay for HIV-1 DNA drug resistance genotyping coupled with a robust bioinformatics pipeline. We report our results of targeted drug resistance genotyping covering the protease, reverse transcriptase and integrase regions.

## 2. Materials and Methods

### 2.1. Evaluation Specimens

Buffy coat samples (BCS) isolated from 148 adult and 18 paediatric whole blood samples received for routine HIV-1 viral load testing with unknown subtypes were selected for this evaluation. All samples were pseudo-anonymised (patient identifying information was replaced with unique numbers), and the panel was created as part of development of an assay for diagnostic purposes under Human Tissue Authority (HTA) license, for the future use on the stored samples in the D3/Penta 21 paediatric randomised controlled trial (NCT04337450). Patient samples with a range of viral loads and expected variation in HIV-1 DNA load were included in this assessment. Nucleic acid isolation was performed on the QIAsymphony (Qiagen, Hilden, Germany) automated platform using the DSP virus/pathogen kit and human whole blood 1000 protocol, as per manufacturer’s instructions.

In total, 166 residual blood samples collected for routine HIV-1 viral load analysis were selected for this evaluation. Sample subsets for analyses included 30 BCS from adults selected for Sanger vs. NGS comparison. NGS evaluation was performed on 148 adult samples (this includes the initial 30 adult samples) and 18 samples from children submitted for HIV-1 testing (Table 1).

### 2.2. Amplification and Sequencing

A nested PCR approach was selected for enhanced sensitivity and specificity. This assay characterises HIV-1 Pol by targeting the protease to reverse transcriptase (PrRT) and integrase (INT) regions separately. PCR reaction mixes contain the following: 1× Qiagen PCR buffer, 2.5 units Qiagen HotStarTaq polymerase, 400 pmol forward and reverse primers, nuclease-free water and 10 µL template. Primers used for first round (locations based on HXB2) PrRT1: GAA GAA ATG ATG ACA GCA TGT CAG GG (1819), PrRT 2: TAA TTT ATC TAC TTG TTC ATT TCC TCC AAT (4173), INT1: TTC TTC CTG CCA TAG GAR ATG CCT AAG (4143), INT2: AGG AGC AGA AAC TTW CTA TGT AGA TGG (5571); for the second round, the following were used: PrRT 3: AGA CAG GCT AAT TTT TTA GGG A (2074), PrRT 4: ATG GYT CTT GAT AAA TTT GAT ATG TCC (3559), INT3: TTC RGG ATY AGA AGT AAA YAT AGT AAC AG (4150), INT4: TCC TGT ATG CAR ACC CCA ATA T (5518). The thermocycling conditions were as follows: 1 cycle at 93 °C for 12 min; 1 cycle at 95 °C for 30 s, 65 °C for 45 s, 72 °C for 3 min; 1 cycle at 95 °C for 30 s, 60 °C for 45 s, 72 °C for 3 min; 45 cycles of 95 °C for 30 s, (58 °C for PCR1 and 55 °C for PCR2) for 45 s, 72 °C for 3 min with a final extension of 10 min at 72 °C. PCR products were analysed with gel electrophoresis for PCR products of ~1.4 kb. Positive amplicon library preparation was performed with Illumina Nextera XT DNA library prep kit (Illumina, San Diego, CA, USA), as per manufacturer’s instructions. Sequencing was performed on an Illumina MiSeq system using v3 600-cycle reagent kits.

### 2.3. Sensitivity

PCR assay sensitivity was evaluated using serially diluted DNA of a quantified 8E5 clonal HIV-1 cell line from 515 copies/µL to below 1 copy/µL. Both PrRT and INT PCR reactions were performed in triplicate at each dilution point. Additionally, real-time-PCR-quantified DNA from a patient sample was also serially diluted (12.7–0.2 copies/µL) to confirm results obtained from 8E5 serial dilution experiments.

We selected a total of 24 BCS, adult (n = 9) and paediatric (n = 15), to determine the HIV-1 DNA assay input thresholds. Human and HIV-1 DNA were quantified according to published methods [22,23]. Furthermore, we determined the total HIV-1 DNA loads compared to human DNA copies of selected samples positive for targeted PrRT and INT amplification and sequencing.

### 2.4. Specificity

Targeted HIV-1 PCR amplification specificity was evaluated using BCS from residual whole blood samples negative for HIV-1 but positive for other bloodborne viruses. This included 7 samples positive for HIV-2, 15 positive for hepatitis B virus (HBV) and 2 for hepatitis C virus (HCV). BCS were processed in the same way as the samples for HIV-1 evaluation.

### 2.5. Reproducibility

Assay reproducibility was investigated in three ways: amplicon replicates from the same DNA sample, library prep replicates from the same amplicon and mixture experiments to confirm minor variant detection, as described below. These assessments were performed for both PrRT and INT targets.

PCR resampling was evaluated with DNA from the 8E5 HIV-1 cell line and a patient sample. The 8E5 HIV-1 cell line was evaluated with duplicate reactions, and 5 replicates of the patient sample were analysed.

Library preparation and sequence reproducibility assessment comprised 12 PrRT and 5 INT samples. Each amplicon was evaluated in duplicate on separate library preparation and sequence runs for reproducibility evaluation.

Discrimination of minor variant detection vs. sequencing errors was addressed computationally. Minor variants were evaluated to a threshold of 2%. High region coverage minimised sequencing error impact. Two samples with known mutations were selected for each PCR target and designated as either major or minor samples. Major-to-minor sample combinations were mixed as follows: 80%/20%, 85%/15%, 90%/10%, 95%/5% and 98%/2%, respectively. This allowed for “minor sample” variation analyses at 20%, 15%, 10%, 5% and 2%.

### 2.6. Bioinformatics Pipeline

Illumina MiSeq data were analysed using an in-house bioinformatics pipeline executed on a high-performance computing cluster. The pipeline uses Trimmomatic v0.40 (Usadel Lab, Dusselforf, Germany) to trim the reads for quality and to remove adapters [24]. Reads are aligned to the human genome using Bowtie 2 v2.5.1 (https://bowtie-bio.sourceforge.net/bowtie2/index.shtml, accessed on 21 August 2023) [25] and filtered to remove host sequences. Reads with hypermutations are detected and removed using a multi-step process: (1) all reads are mapped to HXB2 (GenBank Accession: K03455.1) using Bowtie2 [26], and a consensus sequence is obtained from the alignment using IVAR v1.4.2 (https://github.com/andersen-lab/ivar, accessed on 21 August 2023) [26]; (2) all reads are aligned to the consensus sequence using Bowtie2; (3) pairwise alignments of reads to the consensus sequence are retrieved from the alignment file using sam2fasta (https://sourceforge.net/projects/sam2fasta/files/, accessed on 21 August 2023) and stored as fasta files; and (4) Hypermut2 [27] was used to detect hypermutated reads from the fasta file and reads that Hypermut2 assigns a *p*-value of <0.05 are filtered. Quasitools Hydra (https://github.com/phac-nml/quasitools/, accessed on 21 August 2023) is used to assemble and call variants from the remaining reads. QC metrics, variants and sequences are reported [28].

## 3. Results

### 3.1. Sensitivity

Amplification sensitivity of 8E5 target DNA, performed in triplicate, yielded positive results at both the upper limit of 515.8 c/µL and lower limit of 0.8 c/µL for both PrRT and INT. The amplification sensitivity of the patient sample performed in triplicate yielded positive amplification at starting concentration of 12.7 c/µL down to 0.4 c/µL (Table 2). Results indicate a limit of detection of 0.4 c/µL (4 c/rxn) for both PrRT and INT PCR amplification. We can, therefore, expect consistent results for patient samples with >4 copies of HIV-1 DNA per reaction.

In addition to sensitivity analyses, nine adult and fifteen paediatric samples were used to evaluate the minimum number of HIV-1 DNA copies required for amplification in the presence of human DNA background (Table 3).

Assay DNA input thresholds indicate as little as 0.1 copies/µL (1 copy/reaction) HIV-1 DNA from paediatric samples and 0.46 copies/µL (4.6 copies/reaction) HIV-1 DNA from adult samples could be positively amplified for both PrRT and INT targets in the presence of 9308.4–320,693.1 copies/µL background human DNA (Table 2).

### 3.2. Specificity

Buffy coat cells isolated from samples negative for HIV-1 and positive for other bloodborne viruses (HBV, HIV-2 and HCV) were included for specificity analyses. Samples were selected from a pool of available residual whole blood samples.

Cross-reactivity results indicated one false positive (weak amplification) result out of 24 tested samples (Table 4). This sets the specificity of the HIV-1 target-specific amplification assay at 98%. Any false positive amplification that occurs will subsequently be sequenced, and reactions with no HIV-1 signal will not pass quality control. Sequences from sources other than HIV-1 (including HIV-2) are filtered out by Hydra for not aligning with HXB2.

### 3.3. Reproducibility

Nine mutations were detected in our 8E5 lab strain sequences when compared to the strain found in GenBank (accession: MK115468.1) with identical frequencies across duplicate NGS reactions (Table 5). The two independent PCR and sequencing reactions analysed here have a mean percent frequency variance of 0.0035% across both PrRT and INT targets. Repeat sequencing of our 8E5 lab isolate yielded 100% PrRT and 99.9% INT alignment, confirming no sequence drift. Alignment of our 8E5 isolate to reference MK115468.1 resulted in 96.38% PrRT and 95.31% INT identity, confirming the 8E5 HIV sequence in our positive control/standard. PCR and sequencing reactions performed in duplicate from HIV-1 8E5 cell line control DNA indicate exceptional assay reproducibility.

Assay reproducibility was evaluated for PrRT (12 samples) and INT (5 samples) in duplicate across different library preparations and sequencing runs. The results indicate a reproducibility score of 97.5% for PrRT and 94.9% for INT across all detected mutations (any detected nonsynonymous mutation) at a 5% cut-off (Table 6 and Table 7). Additionally, no discordant drug resistance mutations (DRMs) were detected across this analysis. Amplification and sequencing susceptibility to PCR bias could result in outliers, as observed in Table 6 sample 1 and Table 7 sample 16. However, these outliers did not influence the stability of the overall analyses.

Results from the mixing experiment for minor variant analyses performed at 20–2% minor sample frequency demonstrated a correlation between expected and observed values of mutation frequencies across PrRT and INT regions (Figure 1). The correlation is supported by R2 values of 0.99 for RT up to amino acid position 244 in the PrRT region and 0.97 for the INT region (Table 8). Due to lower R2 values (below 0.97), minor variant mutations detected beyond RT position 244 should be interpreted with care.

For n in [2,5,10,15,20],
FreqMixSample = FreqMajorSample ∗ (1 − n/100) + FreqMinorSample ∗ (n/100)

### 3.4. Evaluation Sample Results

PCR amplification of PrRT and INT targets was conducted on 148 HIV-1-positive samples obtained from adults. Results showed a positive amplification rate of 92% (136/148) for the PrRT target and 95% (141/148) for the INT target, with a 100% sequence success rate. Additionally, of the 18 HIV-1-positive samples from children, 13 yielded successful amplification for the PrRT target (72%) and 15 samples for the INT target (83%). No trends were observed related to HIV-1 DNA concentration and amplification success rate. Mean sequence coverage graphs indicate coverage above 10,000 for the PrRT region and around 8000 for the INT region (Figure 2).

In total, six HIV-1 group M subtypes (A, B, C, D, F and G) and six circulating recombinant forms (CRF) (CRF02_AG, CRF10_CD, CRF01_AE, CRF06_CPX, CRF09_CPX and CRF50_A1D) were successfully detected and sequenced with this Pan-HIV-1 group M assay. Additionally, 144 DRMs in the PrRT and INT regions were successfully detected across all evaluation samples (Table 9), as characterised by the Stanford HIVdb program-sequence analysis tool. Table 9 shows all DRMs detected with MiSeq NGS above 5% frequency across all evaluation samples (e.g., M184IV represents detected M184I and M184V mutations).

### 3.5. Sanger vs. MiSeq Subset

A comparison of 60 reactions between Sanger sequencing and MiSeq NGS demonstrated that NGS consistently detected a significantly higher number of DRMs overall. Sanger sequences were compared to NGS results with a mutation frequency above 5%. Ten discordant DRMs were identified as defined by the Stanford HIVdb program-sequence analysis tool, with 9/10 detected with MiSeq vs. 1/10 on Sanger.

Additionally, the total mutation comparison between NGS and Sanger indicates a concordance of 84.2%, with 96.9% of all mutations detected by NGS (3.1% missed) compared to 87.3% on Sanger (12.7% missed) (Table 10).

## 4. Discussion

There is a growing demand for robust and highly sensitive assays capable of detecting DRMs in HIV-1 DNA. These assays are essential for future studies and clinical trials focused on understanding and targeting viral reservoirs. Furthermore, they could be helpful in clinical practice for treatment optimisation in suppressed patients who require therapy switching. The data presented here demonstrate an evaluation of a sensitive HIV-1 DNA assay for the detection of DRMs in the PrRT and INT regions on the MiSeq NGS platform. Additionally, this study supports the use of BCS, with quicker and easier processing compared to traditional PBMCs.

The evaluation included 166 HIV-1 positive BCS: 148 from adults and 18 from children. The limit of detection evaluation based on the 8E5 control and a patient sample indicates an assay sensitivity of four HIV-1 DNA copies per reaction. Additionally, positive amplification and sequencing were possible for patient samples below this cut-off (down to a single copy per reaction) with background human DNA of up to 318 565 copies per reaction. Therefore, the assay detection threshold is set at four HIV-1 DNA copies per reaction. This level of sensitivity is rarely reported for sequence-based assays [29,30,31].

Sequencing-based assays are inherently less affected by non-specific amplification since off-target sequence results can easily be identified and excluded bioinformatically. Nonetheless, PCR specificity is defined here at 98% with a single HIV-2 positive sample yielding weak amplification, and any non-HIV-1 sequences are filtered out by Hydra as part of the bioinformatics pipeline. Overall amplification and NGS success rate for all evaluated samples were 92% and 99%, respectively, covering six major circulating subtypes and six recombinant forms in HIV-1 group M.

The evaluation of selected positive amplicons in Sanger and NGS platforms showed, as expected [17,18], increased sensitivity for DRMs for NGS, with 9/10 discordant results only detected in NGS sequences, supporting the use of NGS in HIV-1 DNA DRM detection.

The overall NGS sequence coverage for the targeted PrRT and INT regions exceeds 10,000, with one consistent dip in reads within INT to a minimum depth of 8000 reads. The coverage is well above the minimum of 5000 reads set out by one regional regulatory guidance document for submission of NGS data [32]. Assay reproducibility demonstrated consistent and nearly identical results when analysing PCR reactions, library prep, MiSeq sequencing and the bioinformatics pipeline independently. Furthermore, dilution experiments indicate minor variant calling can reliably be performed at the 5% mutation frequency cut-off. Based on all findings in this evaluation, a 5% sequence cut-off was selected for DRM characterisation with high confidence of reproducibility.

This assay was evaluated across eight HIV-1 group M subtypes and six CRFs, indicating a broad range of detection capability. In addition, 144 DRMs were characterised across the PrRT and INT regions, including clinically significant DRMs such as M184IV and R263K. HIV-1 DNA sequencing data interpretation poses specific challenges given that in well-suppressed patients, the majority of viruses in latency are defective and/or are hypermutated. Our approach overcomes some of these challenges, as demonstrated by developing a new bioinformatics pipeline, most significantly to identify and remove hypermutated reads prior to assembly and analyses. The amplification and sensitive variant identification by this assay allow for in-depth HIV-1 DNA analyses and drug resistance genotyping. As the clinical relevance of minor HIV-1 DNA variants is still being explored, we echo the message that this data should be interpreted with care in a clinical setting [17,18].

Here, we present a highly sensitive and specific Pan-HIV-1 assay for PrRT- and INT-targeted drug resistance genotyping from buffy coats. In addition, the robust bioinformatics pipeline is purpose-built and offers reproducible variant and DRM calling. This assay is optimal for monitoring archived HIV-1 drug resistance in HIV-1-suppressed individuals and in clinical trials of novel therapeutic approaches.

## Figures and Tables

**Figure 1 viruses-15-01811-f001:**
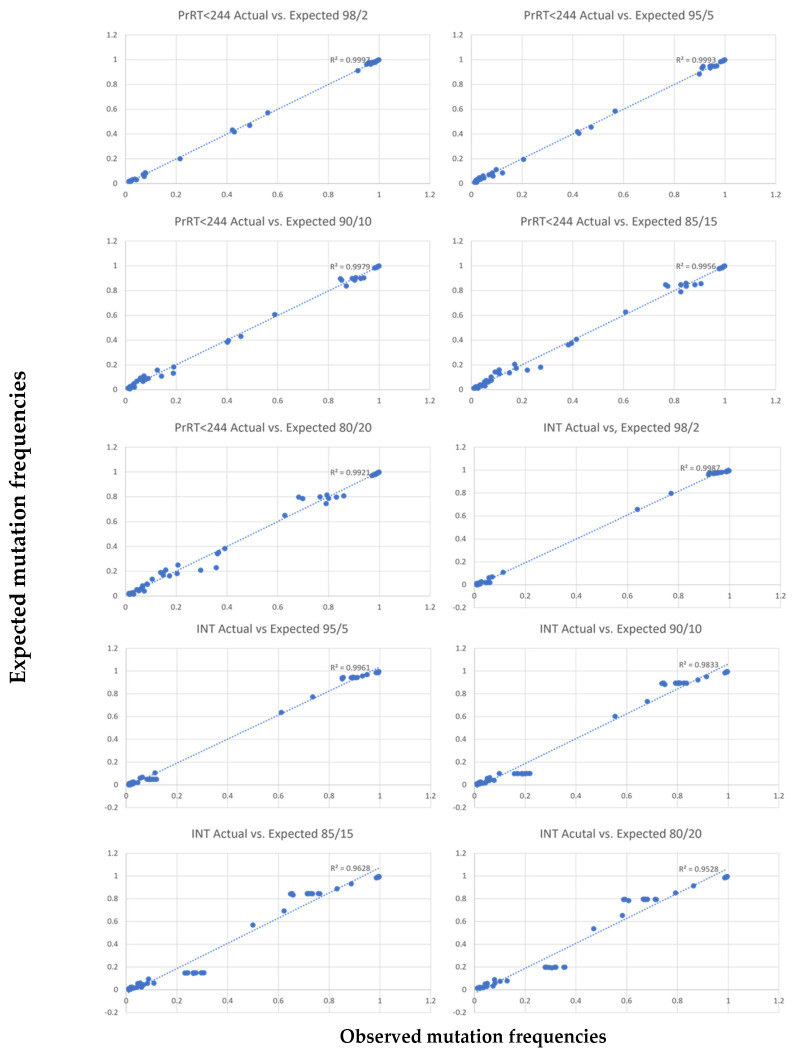
Correlation plots for each major/minor variant mixture, between observed mutation frequencies (*x*-axis) and expected mutation (*y*-axis) frequencies (5% cut-off applied). The expected frequency is calculated.

**Figure 2 viruses-15-01811-f002:**
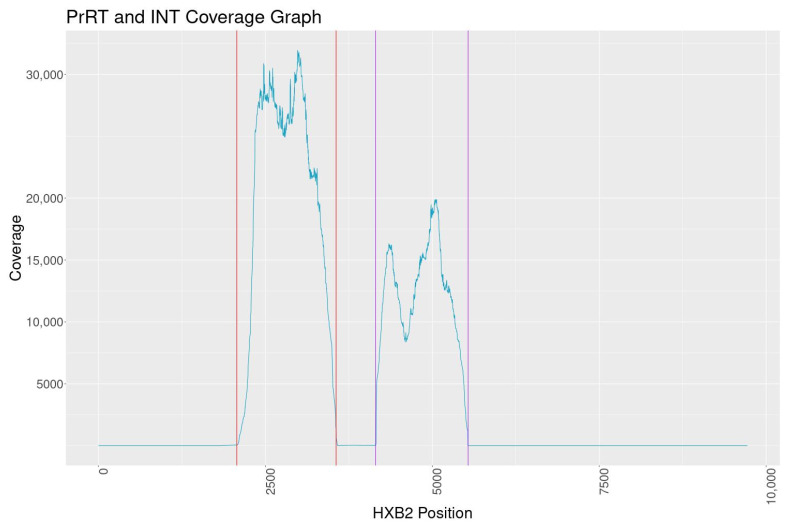
Mean sequence coverage graph of PrRT and INT MiSeq NGS vs. HXB2 (Genbank: K03455.1). The coloured vertical lines represent primer binding locations.

**Table 1 viruses-15-01811-t001:** Study participant characteristics.

Characteristic	30 Adult Samples Subset	All Adult Samples	Paediatric Samples
**Female**	12	65	4
**Male**	18	83	14
**Age range (median), years**	23–78 (50.5)	17–78 (49)	2–17 (14.5)
**HIV-1 (<50 c/mL)**	14	89	16
**HIV-1 (50–1000 c/mL)**	3	17	2 (<150 c/mL)
**HIV-1 (1000–10 000 c/mL)**	2	11	None
**HIV-1 ≥10 000 c/mL**	11	31	None
**Total samples**	**30**	**148**	**18**

**Table 2 viruses-15-01811-t002:** Limit of detection analyses on serially diluted DNA in triplicate.

Sample	Copies/µL	Copies/Reaction	PrRT Positive Reactions	INT Positive Reactions
**8E5 cell line DNA**	515.8	5157.6	3	3
	103.2	1031.5	3	3
	51.6	515.8	3	3
	25.8	257.9	3	3
	12.9	128.9	3	3
	6.4	64.5	3	3
	3.2	32.2	3	3
	1.6	16.1	2 *	2 *
	0.8	8.1	3	3
**Study** **sample**	12.7	127.0	3	3
	6.35	63.5	2 *	3
	3.18	31.8	3	3
	1.59	15.9	3	3
	0.79	7.9	3	3
	0.4	4.0	3	3
	0.2	2.0	0	1

* Serial dilutions at low DNA concentrations risk exclusion of target DNA from some aliquots.

**Table 3 viruses-15-01811-t003:** Human and HIV-1 DNA copy numbers required for positive target amplification.

Sample Type	Sample ID	Copies/µL		Targeted HIV-1 PCR Positive	
		Human DNA	HIV-1 DNA	PrRT	INT
**Adults**	ID_37	24,112.4	0.46	Positive	Positive
	ID_32	202,714.3	0.65	Positive	Positive
	ID_33	165,062.9	2.01	Positive	Positive
	ID_36	163,217.1	2.27	Positive	Positive
	ID_30	297,115.4	4.16	Positive	Positive
	ID_34	117,258.8	6.72	Positive	Positive
	ID_35	162,380.1	8.05	Positive	Positive
	ID_38	320,693.1	10.42	Positive	Positive
	ID_29	172,073.2	12.7	Positive	Positive
**Children**	ID_C_16	10,561.8	<0.1 *	Negative	Positive
	ID_C_25	18,334.3	<0.1 *	Positive	Positive
	ID_C_40	31,856.5	<0.1 *	Positive	Positive
	ID_C_32	23,190.2	0.1	Positive	Positive
	ID_C_28	19,819.5	0.2	Positive	Positive
	ID_C_4	26,576.4	0.2	Positive	Positive
	ID_C_29	20,857.2	0.3	Negative	Positive
	ID_C_2	20,314.4	0.5	Positive	Positive
	ID_C_31	12,519.1	0.7	Positive	Positive
	ID_C_17	9308.4	0.8	Positive	Positive
	ID_C_38	16,371.8	1.1	Positive	Positive
	ID_C_24	12,138.6	1.6	Positive	Positive
	ID_C_33	20,367.2	1.7	Negative	Positive
	ID_C_15	22,215	1.9	Positive	Positive
	ID_C_9	17,076.1	12	Negative	Negative

* Less than 0.1 c/µL of HIV-1 DNA detected in quantification results equates to a single copy per reaction.

**Table 4 viruses-15-01811-t004:** Samples for assay specificity evaluation.

Target Sample	Tested	False Positive	
		PrRT	INT
**HBV**	15	0	0
**HIV-2**	7	1	0
**HCV**	2	0	0
**Total**	24	1	0

**Table 5 viruses-15-01811-t005:** HIV-1 8E5 cell line control comparison from two independent PCR and sequencing reactions for each target.

Region	Mutations Compared to 8E5 (MK115468.1)	Frequency of 8E5-1	Frequency of 8E5-2
**PrRT**	V3I	0.99	0.99
	L214F	0.99	0.99
	M357T	0.99	0.99
	K388R	0.99	0.99
**INT**	N232D	0.99	0.99
	R127K	0.99	0.99
	G123S	0.99	0.99
	A265V	0.99	0.99
	A124T	0.99	0.99

**Table 6 viruses-15-01811-t006:** PrRT reproducibility results (% of uniquely identified sequences) across two runs for frequency of mutations detected at ≥5%.

Sample	Run 1 Only	Both Runs	Run 2 Only
1	13.5	86.5	0
2	0	98.2	1.8
3	0	100	0
4	0	100	0
5	5.3	92.1	2.6
6	0	100	0
7	0	100	0
8	0	96	4.1
9	0	100	0
10	0	100	0
11	0	100	0
12	2.4	97.6	0
**Average**	**1.8**	**97.5**	**0.7**

**Table 7 viruses-15-01811-t007:** INT reproducibility results (% of uniquely identified sequences) across two runs for frequency of mutations detected at ≥5%.

Sample	Run 1 Only	Both Runs	Run 2 Only
13	0	97.4	2.9
14	0	94	6
15	0	100	0
16	11.1	83.3	5.6
17	0	100	0
**Average**	**2.2**	**94.9**	**2.9**

**Table 8 viruses-15-01811-t008:** R2 values when comparing observed and expected frequencies.

Dilution	PrRT	Pr	RT	* RT < 244	* PrRT < 244	INT
**2%**	0.9825	0.9997	0.9791	0.9998	0.9992	0.9987
**5%**	0.9424	0.9995	0.9262	0.9992	0.9993	0.9961
**10%**	0.8802	0.9984	0.839	0.9978	0.9979	0.9833
**15%**	0.8108	0.9974	0.7354	0.9949	0.9956	0.9628
**20%**	0.7491	0.997	0.6436	0.989	0.9921	0.9528

* RT < 244 and PrRT < 244 show R2 values when considering AA position 244 or less in RT.

**Table 9 viruses-15-01811-t009:** Overview of DRMs detected with MiSeq NGS across all samples at frequencies ≥5%.

PI Major	PI Accessory	NRTI	NNRTI	INSTI Major	INSTI Accessory
D30N	L10FV	M41L	A98G	T66AIKMV	H51Y
M46IL	L33F	A62V	L100EI	E92AGKR	L74F
I47V	K43T	K65R	K103ENRS	G118RS	L74IMT
I50L	I47M	D67AGHN	V106I	E138AK	T97A
I54L	G48ER	S68NGR	V108I	G140EKRS	A128T
V82AIT	Q58E	K70EINRS	E138AGKQ	Y143CHR	G140E
N88S	G73DRS	V75I	V179DE	Q146R	P145L
L90M	T74P	M184IV	Y181CS	S147GN	S153A
	N83D	L210W	Y188CFHL	Q148R	E157Q
	L89MV	T215ADN	G190EKRS	N155DH	E157KQ
		T215CFSY	H221Y	R263K	G163AEKRTS
		T215NSY	F227L		D232N
		K219EHKNQR	M230I		
			K238T		
			Y318F		
			N348I		

**Table 10 viruses-15-01811-t010:** All mutations detected in Sanger compared to MiSeq for all samples.

	MiSeq Only	Both	Sanger Only
Number	409	2717	101
Percentage	12.7%	84.2%	3.1%

## Data Availability

The data are held at UCLH APDU, which encourages optimal use of data by employing a controlled access approach to data sharing, incorporating a transparent and robust system to review requests and providing secure data access consistent with the relevant ethics committee approvals. We will consider all requests for data sharing, which can be initiated by contacting Eleni Nastouli.

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
