# Peer review of "Sensitive HIV-1 DNA Pol Next-Generation Sequencing for the Characterisation of Archived Antiretroviral Drug Resistance"

_viruses, 2023, doi:10.3390/v15091811_

Round 1
Reviewer 1 Report
This manuscript is focused on evaluation of an NGS assay for HIV-1 drug resistance on BCS. The significance of HIV DR from HIV reservoir might be underestimated because of having not enough data. Thus it is important to establish reliable and comparable methods. The authors evaluated the established assay from sensitivity, specificity, and reproducibility, which are reasonable. However, there are some points to be refined: e.g. the position of primers, how long of the amplicons, possible reasons for differences of sequences between NGS and original copy of 8E5, how the DNA load of samples in tables 6 and 7, possible reasons for outliers from samples 1 and 16.
Author Response
We’d like to thank the editor and reviewers for their time and patience reviewing our work, thank you for your comments and suggestions. Please see the edits and responses below.
Reviewer 1
This manuscript is focused on evaluation of an NGS assay for HIV-1 drug resistance on BCS. The significance of HIV DR from HIV reservoir might be underestimated because of having not enough data. Thus it is important to establish reliable and comparable methods. The authors evaluated the established assay from sensitivity, specificity, and reproducibility, which are reasonable.
However, there are some points to be refined: e.g. the position of primers, how long of the amplicons,
We’ve added primer locations based on HIV HXB2. Lines 80-86.
possible reasons for differences of sequences between NGS and original copy of 8E5. We observed 100% (PrRT) and 99% (INT) alignment identity compared to the specific strained we received in our lab. The two published 8E5 strains MK115468.1 and MN090435.1 shows only 89.10% identity when compared to each other (we did not include this analysis as it does not fall within the scope of this paper).
, how the DNA load of samples in tables 6 and 7. This library preparation and sequencing comparison was performed from the same amplicons, because of this DNA loads are not indicated.
, possible reasons for outliers from samples 1 and 16. We performed extensive assay reproducibility analyses with good results. Library preparation does include an amplification step that can in some instances result in biased amplification which is unavoidable with current methods.
Reviewer 2 Report
The manuscript of Botha et al described a sensitive HIV-DNA pol next generation sequencing method on Miseq platform from buffy coat samples. There is few data available to date to my knowledge in this field however the results are not clearly presented and some tables are not necessary.
The authors prone a sensitive methods but do not described the sensitivity of NGS HIV-1 DNA available to date and did not discussed their sensitivity compared to published data so the contribution of their data is not persuassive.
1/ Introduction
The introduction do not sufficiently introduce the sensitivity of actuel HIV-1 DNA pol NGS techniques already available.
2/ Results :
2.1 Could you please express HIV-1 DNA viral load in copies by million of cells which is the unit commonly used ? (in Table 2, Table 3…).
2.2 It is quite surprising to observed successful amplification of PrRT and INT in ID_C16, ID_C_25 and ID_C_40 has HIV-1 DNA is <0.1 c/µL. How can you explain these results ? In Table 3, it would be interesting to add the RNA viral load. It would also be interesting to do phylogenetic analysis to check there is no cross-sample contamination, especially for samples with low HIV-1 DNA viral load.
2.3 Table 4 can be removed simply describing data in the text.
2.4 Table 5 is not clear modified the presentation with 2 column (one for each independent PCR and sequencing ex : frequency 1 and frequency 2).
2.5 Table 6 and 7 are not understandables : if frequency of mutation are listed in the table, please explain it and list the mutations detected. The term reproducibility is used but the column « both runs » show data representing the inverse frequency of the one obtained in « run 1 only » or « run 2 only » or both columns.
2.6 Figure 1 is not well explained, please described the horizontal and vertical axis in the figures and not in the legends. Descriptions in lines 233-235 should be included in the results section. I’m not certain the figure
2.7 Does HIV subtypes were known before NGS results ? if so described how and include the subtypes data in Table 1.
2.8 Tables 9 and 10 are useless. The same for Table 11 were data are presented in the text.
Author Response
We’d like to thank the editor and reviewers for their time and patience reviewing our work, thank you for your comments and suggestions. Please see the edits and responses below.
Reviewer 2
The manuscript of Botha et al described a sensitive HIV-DNA pol next generation sequencing method on Miseq platform from buffy coat samples. There is few data available to date to my knowledge in this field however the results are not clearly presented and some tables are not necessary.
The authors prone a sensitive methods but do not described the sensitivity of NGS HIV-1 DNA available to date and did not discussed their sensitivity compared to published data so the contribution of their data is not persuassive.
1/ Introduction
The introduction do not sufficiently introduce the sensitivity of actuel HIV-1 DNA pol NGS techniques already available. We’ve included references 10-12 and 15-17 including “Parikh, U.M.; McCormick, K.; van Zyl, G.; Mellors, J.W. Future Technologies for Monitoring HIV Drug Resistance and Cure. Curr Opin HIV AIDS 2017, 12, 182–189” to highlight some of the available methods.
2/ Results :
2.1 Could you please express HIV-1 DNA viral load in copies by million of cells which is the unit commonly used ? (in Table 2, Table 3…). Cell numbers from buffy coats were not counted, because this would add an additional step to the method and it was not necessary to determine copy numbers per 10^6 cells as we have shown. DNA copies per PCR reaction was used because this highlights better the sensitivity of the assay and shows that the method works over a wide range of HIV copy numbers. Therefore, cells/1M standardisation of sample input is not necessary here.
2.2 It is quite surprising to observed successful amplification of PrRT and INT in ID_C16, ID_C_25 and ID_C_40 has HIV-1 DNA is <0.1 c/µL. How can you explain these results ? The sample input per reaction is 10ul (included in line 80), therefore, 0.1 c/ul equals to 1 copy per reaction (added to table 3). We show in our manuscript that our method is very sensitive and can even detect one HIV copy per reaction.
In Table 3, it would be interesting to add the RNA viral load. It would also be interesting to do phylogenetic analysis to check there is no cross-sample contamination, especially for samples with low HIV-1 DNA viral load. We harvest the cells from buffy coat, which excludes plasma RNA. Therefore, the VL is not relevant for the method and we did not include it. Furthermore, there is no correlation between cell associated DNA and plasma VL.
2.3 Table 4 can be removed simply describing data in the text. We feel that Table 4 summarises the results in a more graphic way which could be easier to comprehend for some readers. Therefore, we would like to leave it in (or move to Supp. if Editor advises).
2.4 Table 5 is not clear modified the presentation with 2 column (one for each independent PCR and sequencing ex : frequency 1 and frequency 2). We’ve amended the table aesthetically to aid clarity.
2.5 Table 6 and 7 are not understandables : if frequency of mutation are listed in the table, please explain it and list the mutations detected. The term reproducibility is used but the column « both runs » show data representing the inverse frequency of the one obtained in « run 1 only » or « run 2 only » or both columns. We’ve added (% of uniquely identified sequences) to table headings to clarify. This evaluation includes all detected mutations (not only drug resistance relevant mutations) this is into the 100s and not relevant or practical to display.
2.6 Figure 1 is not well explained, please described the horizontal and vertical axis in the figures and not in the legends. Descriptions in lines 233-235 should be included in the results section. I’m not certain the figure. Added x and y axis descriptions to figure 1.
2.7 Does HIV subtypes were known before NGS results ? if so described how and include the subtypes data in Table 1. Added “with unknown subtypes” to line 58.
2.8 Tables 9 and 10 are useless. The same for Table 11 were data are presented in the text. Table 11 could be moved to Supp data. We feel that Tables 9 and 10 summarises the results in a more graphic way which could be easier to comprehend for some readers. Therefore, we would like to leave it in (or move to Supp. if Editor advises). Many clinical readers would appreciate the inclusion of all detected drug resistance mutations.
Reviewer 3 Report
Provide more information about 8E6.
Only a few minor comments:
What is new? NGS is already recommended by other authors. Did other authors use BCS for NGS?
…….total of 24 adult (n=9); 24 samples from 9 individuals?
In Table 10, can you indicate which ones were not detected by Sanger and vice versa?
OK
Author Response
We’d like to thank the editor and reviewers for their time and patience reviewing our work, thank you for your comments and suggestions. Please see the edits and responses below.
Provide more information about 8E6. HIV strain is widely used as a HIV assay control, we’ve included reference “Busby, E.; Whale, A.S.; Ferns, R.B.; Grant, P.R.; Morley, G.; Campbell, J.; Foy, C.A.; Nastouli, E.; Huggett, J.F.; Garson, J.A. Instability of 8E5 Calibration Standard Revealed by Digital PCR Risks Inaccurate Quantification of HIV DNA in Clinical Samples by QPCR. Sci Rep 2017, 7, 1209, doi:10.1038/s41598-017-01221-5.” For this purpose.
Only a few minor comments:
What is new? NGS is already recommended by other authors. Did other authors use BCS for NGS? Added “from buffy coats” line 325
…….total of 24 adult (n=9); 24 samples from 9 individuals? Corrected in text.
In Table 10, can you indicate which ones were not detected by Sanger and vice versa? This is data from the entire MiSeq evaluation and not just the Sanger vs MiSeq subset. Added “DRMs detected with MiSeq NGS” for clarity in line 259 also reflected in Table 9 title.
Author Response
We’d like to thank the editor and reviewers for their time and patience reviewing our work, thank you for your comments and suggestions. Please see the edits and responses below.
Viruses-2488223 This article describes the validation and implementation potential of a DNA NGS assay to characterize archived DRMs. As alluded by the authors, this technology could be very useful for reservoir research. The manuscript is well written and data is clearly presented.
I would however like to make a few suggestions:
- Please clarify pseudo-anonymised. Added “patient identifying information replaced with unique numbers” lines 59-60
- Please clarify line 204-205: are “all detected mutations” referring to any amino acid differences from the HX2B reference, including drug resistance mutations? Clarified in text “any detected nonsynonymous mutation” lines 209-210
- How was the cut-off at RT 244 determined for the R2 values? Was this based on the position of clinically relevant mutations, or was this determined based on the optimal R2 that was obtained? Clarified cut-off of 0.97 in line 226.
- Very high PCR amplification success rates were observed among patient samples. For those samples that failed to amplify, was there any trend in a lower HIV-1 DNA concentration in these samples? Added “No trends were observed related to HIV-1 DNA concentration and amplification success rate.” Lines 246-247.
- For the NGS-Sanger comparison, was Sanger sequencing also obtained from DNA, or from RNA? Sequenced from the same amplicons obtained from DNA. Clarified buffy coat samples in line 68.
- Is it fair to compare NGS results at a 5% cut-off with Sanger data which is known to have a cut-off at 15-20%? What did the differences look like when the comparison was done at 20%? We aim to demonstrate the best obtainable results on the platform therefore, evaluating the platforms to each respective relevant cut-off.
- Table 9: this info could easily be incorporated in the text, to reduce the number of tables. Data incorporated into text and table was removed. Lines 254-256.
- Table 5 and 10 could be moved to supplementary data. We feel that this is valuable data. Once again, we could move this to Supp data if the Editor advises.
Round 2
Reviewer 2 Report
1/ In the introduction, the sensitivity of actuel HIV-1 DNA pol NGS techniques is not well described. In fact, the references listed are often review which did not precise the sensitivity of their methods. Please clearly indicate a range of sensitivity of already available methods. This will give weigth to your stud with described an ultrasensitive method.
2/ For Table 5, the author’s reponse is « We’ve amended the table aesthetically to aid clarity. », I recommend to modified the Table as suggesting in my previous reviewing for clarity reasons introducing a column with « Frequency of 8E5-1 » and another for « Frequency of 8E5-2 » and replacing column « sample » by « Regions » (ex : PrRT and INT).
3/In figure 1, please replace the term « Actual » by « Observed » which is more appropriate.
Author Response
We’d like to thank the editor and reviewers for their time and patience reviewing our work, thank you for your comments and suggestions. Please see the edits and responses below.
General edits:
Some small edits were made, all marked in in yellow.
1/ In the introduction, the sensitivity of actuel HIV-1 DNA pol NGS techniques is not well described. In fact, the references listed are often review which did not precise the sensitivity of their methods. Please clearly indicate a range of sensitivity of already available methods. This will give weigth to your stud with described an ultrasensitive method.
Added text and references: Several DNA HIVDR NGS based assays are available with the most prominent clinically used assays reporting amplification success rates of 58.4% to 94% from purified peripheral blood mononuclear cells (PBMCs) [13,14]. Other studies reported HIV-1 amplification sensitivity as low as 10 and 131 DNA copies per reaction [15,16].
2/ For Table 5, the author’s reponse is « We’ve amended the table aesthetically to aid clarity. », I recommend to modified the Table as suggesting in my previous reviewing for clarity reasons introducing a column with « Frequency of 8E5-1 » and another for « Frequency of 8E5-2 » and replacing column « sample » by « Regions » (ex : PrRT and INT).
Table 5 was modified as requested
3/In figure 1, please replace the term « Actual » by « Observed » which is more appropriate.
Done in figure and legend.